# X-ray and Thermal Analysis of Selected Drugs Containing Acetaminophen

**DOI:** 10.3390/molecules25245909

**Published:** 2020-12-13

**Authors:** Izabela Jendrzejewska, Tomasz Goryczka, Ewa Pietrasik, Joanna Klimontko, Josef Jampilek

**Affiliations:** 1Institute of Chemistry, University of Silesia, Szkolna 9, 40007 Katowice, Poland; ewa.pietrasik@us.edu.pl; 2Institute of Materials Science, University of Silesia, Bankowa 12, 40007 Katowice, Poland; tomasz.goryczka@us.edu.pl; 3Institute of Physics, University of Silesia, Uniwesytecka 4, 40007 Katowice, Poland; joanna.klimontko@us.edu.pl; 4Institute of Neuroimmunology, Slovak Academy of Sciences, Dubravska cesta 9, 84510 Bratislava, Slovakia; josef.jampilek@gmail.com

**Keywords:** paracetamol, drugs, X-ray diffraction, phase analysis, thermal analysis

## Abstract

Studies carried out by X-ray and thermal analysis confirmed that acetaminophen (paracetamol), declared by the manufacturers as an Active Pharmaceutical Ingredient (API), was present in all studied medicinal drugs. Positions of diffraction lines (2θ angles) of the studied drugs were consistent with standards for acetaminophen, available in the ICDD PDF database Release 2008. |Δ2θ| values were lower than 0.2°, confirming the authenticity of the studied drugs. Also, the values of interplanar distances *d_hkl_* for the examined samples were consistent with those present in the ICDD. Presence of acetaminophen crystalising in the monoclinic system (form I) was confirmed. Various line intensities for API were observed in the obtained diffraction patterns, indicating presence of the preferred orientation of the crystallites in the examined samples. Thermal analysis of the studied substances confirmed the results obtained by X-ray analysis. Drugs containing only acetaminophen as an API have melting point close to that of pure acetaminophen. It was found that presence of other active and auxiliary substances affected the shapes and positions of endothermal peaks significantly. A broadening of endothermal peaks and their shift towards lower temperatures were observed accompanying an increase in the contents of additional substances being “impurities” in relation to the API. The results obtained by a combination of the two methods, X-ray powder diffraction (XRPD) and differential scanning calorimetry/thermogravimetry (DSC/TGA), may be useful in determination of abnormalities which can occur in pharmaceutical preparations, e.g., for distinguishing original drugs and forged products, detection of the presence of a proper polymorphic form or too low content of the active substance in the investigated drug.

## 1. Introduction

Self-medication has become very popular in recent times, and drug manufacturers are constantly striving to make more and more drugs available over the counter. Such drugs include acetaminophen (ACP, paracetamol), belonging to *p*-aminophenol derivatives, having a formula CH_3_CONHC_6_H_4_OH (C_8_H_9_NO_2_). Despite its simple structure, ACP occurs in two polymorphic variations: monoclinic (form I) and orthorhombic (form II) [1]. At room temperature, form I is more thermodynamically stable than form II [2]. This compound is included in the composition of more than 300 pharmaceutical preparations. Advantages of ACP include rapid action, high effectiveness, low toxicity and very rare occurrence of side effects. Due to these facts, ACP is one of the most frequently used analgesics and antipyretics. It is available over the counter not only in pharmacies, but also in shops, gas stations, or from the Internet. ACP is marketed under various trade names, as a sole active pharmaceutical ingredient (API) or combined with other APIs, enhancing its power and scope of effect (caffeine, codeine, propyphenazone, pseudoephedrine, acetylsalicylic acid (ASA), ascorbic acid). It is available in the form of pills, capsules, effervescent tablets, syrups or powder for preparation of solutions.

Mass counterfeiting of pharmaceutical products has become the plague of the 21st century. World Health Organization (WHO) and U.S. Food and Drug Administration (FDA) experts estimate that the most frequently forged medicinal drugs include: erectile dysfunction drugs (Viagra, Cialis), generic antibiotics (28%), analgesics, antiasthmatic and antiallergic drugs (8%), medications against AIDS/HIV, hormones (18%), antimalarial drugs (7%), as well as drugs used in anxiety disorders (Xanax, Ativan) [3,4]. The counterfeiters are also interested in popular medicines, such as aspirin [5] or ACP. In 2019, counterfeit antineoplastic drugs were discovered in Canada, containing nothing apart from ACP, available over the counter [6]. In 2020, Canada issued a warning that tablets appearing like prescription oxycodone or Percocet were available on the market. The public was warned that fentanyl may be present in these tablets. Fentanyl is a strong opioid and even in a small amount may cause an overdose or death [7]. In 2018, counterfeit paracetamol tablets being sold under the trademark “Biogesic” were discovered [8].

Counterfeit medicines are made all over the world. Packaging and medicines are often produced and printed in different countries and then sent to their final destination, where they are collected and distributed. India, China, Vietnam, Indonesia, Pakistan and the Philippines appear to be major producers of counterfeit medicines traded worldwide, and Hong Kong, the United Arab Emirates, Egypt, Cameroon and Turkey appear to be major transit points for counterfeit medicines and pharmaceuticals shipped worldwide [9]. Counterfeit pharmaceutical preparations pose a danger for human life and health, therefore it is important to monitor and analyse pharmaceutical materials. The majority of drugs is constituted by crystalline substances in solid state, so commonly used techniques may be applied for this, such as X-ray diffraction, thermal analysis, infrared spectroscopy (IR) or various types of microscopy. A combination of two or more investigation methods allows for broadening of scope of the studied properties, as well as confirming the authenticity and composition of the examined drug. Classical separation methodologies, such as chromatography or electrophoresis (e.g., for ACP [10,11,12,13]) can also be used, but there it is necessary to process the samples (unlike near IR or Raman spectrometry) and the solid phase information is lost upon dissolution.

### 1.1. X-ray Structural Analysis

Diffraction occurring on materials with crystalline structure is one of the fundamental tools used for characterisation of substances. Reflection of X-ray radiation (or electromagnetic waves with wavelengths comparable to interatomic distances) occurs on a set of parallel lattice planes (*hkl*). The radiation will be amplified when the angle of incidence (θ) is equal to the angle of reflection (θ), i.e., when the difference between their optical paths is equal to a total multiple of the wavelength, that is when the waves are in phase (Figure 1).

The amplification condition will be met if
*n*λ = 2*d_hkl_* sinθ(1)
where: *n*—reflection order, λ—wavelength, *d_hkl_*—interplanar distance, θ—angle of reflection. Equation (1) is the Wulff-Bragg equation. It is a basic equation describing the geometrical condition for X-ray diffraction on lattice planes having interplanar distances of *d_hkl_*.

In order to characterize a material fully, the identification of the crystalline phases occurring in it is necessary. Every polycrystalline phase has a specific X-ray photograph with characteristic positions and intensities of diffraction lines. Such polycrystalline diffraction patterns are so complex that it is not possible for different substances to have identical diffraction, therefore, such patterns may play the role of “fingerprints” in the identification of substances. Every polycrystalline phase present in a phase mixture yields a specific X-ray photograph, irrespective of the phases which co-exist with it. As the *d_hkl_* values do not depend on the wavelength used in the experiment or its geometry, such an X-ray photograph may be also presented as a set of interplanar distance *d_hkl_* and intensities of diffraction lines corresponding to them.

The presence of diffraction lines characteristic of a given phase confirms its presence in the examined material. Lack of diffraction lines of the phase being sought for does not mean its absence in the tested material, but may indicate that its amount is lower than its X-ray limit of detection (lowest percentage of a polycrystalline substance (phase) in analysed material, below which its diffraction pattern ceases to be recorded). The limit of detection of crystalline phases in diffractometry is in the range of 0.1–1% by wt. and depends on the crystal structure of a given phase, characters of co-existing phases, and instrumental factors. It is assumed that the limit of detection (LOD) amounts to approx. 1% [14,15].

X-ray photographs of phases having a high symmetry (regular, tetragonal, hexagonal systems) usually contain a low number of diffraction reflexes, but of high intensities. It allows for identifying them at their contents below 1% by wt. Phases composed of atoms of heavy atoms will exhibit higher intensities of reflexes than those composed of light elements.

Qualitative phase analysis consists in comparing the obtained diffraction pattern (2θ angles, interplanar distances *d_hkl_*, and intensities of diffraction lines) with a proper standard available in a diffraction database. A shift of diffraction lines by less than 0.2° is a normal phenomenon during an analysis of a polycrystalline substance, caused by random arrangement of grains in a polycrystalline sample. On the other hand, shifts larger than 0.2° at a given diffraction angle 2θ will indicate a presence of a different crystal structure. Considering the above, a product for which the obtained diffraction lines are shifted by more than 0.2°, should be considered suspect [16,17]. This rule is published in the general chapter of the USP <941> ascertaining that if the shifts of diffraction lines in an XRD image of the tested products are larger than 0.2° for a given 2θ diffraction angle while compared to the XRD image for an authentic product, these products meet the counterfeit criteria [18,19].

Owing to this rule, counterfeit medical products, drugs or dietary supplements, may be distinguished from authentic materials based on their diffraction patterns. Additional lines, lack of lines, as well as line shifts will be observed in diffraction patterns of counterfeit products. Qualitative phase analysis may be used successfully as technique allowing for distinguishing of authentic pharmaceutics from counterfeit products [20,21].

### 1.2. Thermal Analysis

Thermal analysis methods provide information on changes in selected substance properties under the influence of a specific temperature change. In the case of drugs, these methods are used to determine such thermal parameters as melting point, decomposition point, stability, compatibility, polymorphic transformations or interactions between drugs and drugs/excipients. These properties often determine the quality of the product. The advantages of thermogravimetric analysis (TGA) and differential scanning calorimetry (DSC) rely on the fast sample measurement, a small amount of sample required, and easy detection of physical properties [22,23].

In our studies, a combination of X-ray phase analysis based on polycrystalline diffraction (X-ray powder diffraction–XRPD) and DSC/TGA thermal analysis (thermogravimetric analysis–TGA) to check the presence of ACP in drugs stated to contain this compound as an API. Popular and widely available drugs were chosen for the research. The goal of the paper consisted of identification of ACP in selected drugs. Additionally, we aimed for indicating the possibility and effectiveness of a combination of such two research methods as X-ray analysis and thermal analysis in the determination of authenticity of a product. Owing to the above, one may detect whether the API in question is present in the tested drug and what is its crystalline form (polymorphic form).

## 2. Results and Discussion

### 2.1. X-ray Phase Analysis

Qualitative phase analysis used a comparison of experimental diffraction data such as 2θ diffraction angles, *d_hkl_* interplanar distances, and relative intensities, with the data from the ICDD database. Values of *d_hkl_* interplanar distances were calculated based on the Bragg-Wulff equation. Table 1 and Figure 2 show the results of XRPD analysis for pure ACP. On the other hand, the results obtained for drugs containing ACP are grouped according to the drug composition, considering the absence of an additional API (Figure 3 and Figure 4, Table 2 and Table 3) or its presence (e.g., caffeine, ASA) in the tested drugs (Figure 5, Figure 6, Figure 7, Figure 8 and Figure 9, Table 4, Table 5, Table 6, Table 7 and Table 8). In the diffraction patterns, all diffraction lines originating from ACP are indicated; meanwhile, the comparison of experimental diffraction data with those from the International Centre of Diffraction Data Powder Diffraction Files 2 (ICDD PDF2) 2008 Release 2008 has been carried out for the strongest three diffraction lines of ACP [24].

#### 2.1.1. X-ray Characteristic of Pure ACP

Analysis of a diffraction pattern of pure ACP proved very good accordance with the PDF card 00–039–1503 (Figure 2, Table 1). Crystallographic parameters were determined by LeBail method: monoclinic system, SG P2_1_/n, a = 11.7458(5) Å, b = 9.4187(2) Å, c = 7.1363(5) Å, β = 97.521°. Deformed diffraction lines and their higher intensities while compared to the ICDD data (Table 1) indicate presence of a preferred orientation of crystallites in the tested sample [25].

The relationship between the intensity of any diffraction line (reflex) $J_{hkl}^{j,0}$ for phase *j* having mass absorption coefficient $\mu_{j}^{*}$, and the intensity of the same reflex $J_{hkl}^{j}$ for phase *j* present in a polyphase mixture (mass share of phase *j* in the polyphase mixture is expressed as $m_{j}$, and its mass absorption coefficient is equal to $\mu^{*}$) is expressed by Equation (2):(2)$$J_{hkl}^{j}=J_{hkl}^{j,0}\frac{\mu_{j}^{*}}{\mu^{*}}m_{j}$$

The shape of the diffraction line depends on the content of the given phase in the mixture, crystal structure of the former, the character of co-existing phases, and the crystallite orientations [14]. In a case when a privileged crystallite orientation exists, a change in intensities of the individual diffraction lines is observed depending on the direction of dispersion (2θ angle). The experimental values for pure ACP remain in a very good accordance with the data from the ICDD database. The strongest five diffraction lines of ACP are compared (Table 1).

#### 2.1.2. X-ray Phase Analysis of Drugs Containing Only ACP

Figure 3 and Figure 4 present polycrystalline diffraction patterns of drugs containing only ACP as the API. In all diffraction patterns, diffraction lines characteristic for ACP (C_8_H_9_NO_2_) are present. The position of the diffraction lines (2θ angles) and their intensities for Paracetamol Biofarm, Paracetamol Aflofarm, Acenol, and Apap USP are consistent with the data in the ACP PDF card 00–054–2055. A comparison of the experimental diffraction data for the drugs with the ICDD data is shown in Table 2. |Δ2θ| values are lower than 0.2°, confirming presence of ACP in these drugs. Only deviations in intensities of diffraction lines are observed. The experimental values are higher than those from the ICDD database, indicating a preferred orientation of grains in the tested samples. The calculated values of interplanar distances *d_hkl_* have values close to those from the ICDD database (Table 2).

In the case of Paracetamol Polfa and Paracetamol Apteo, it was found that positions of diffraction lines (2θ) in the diffraction patterns and their intensities are consistent with the data for ACP contained in the PDF 00–027–1902 card (Figure 4). Although, the intensities of diffraction lines of Paracetamol Polfa indicate a preferred orientation of crystallites in this sample.

The determined values of |Δ2θ| are lower than 0.2°, confirming presence of ACP in the tested drugs. A comparison of diffraction line intensities does not indicate any preferred orientation of grains in the examined samples. The determined values of interplanar distances *d_hkl_* remain in a very good accordance with the *d_hkl_* values from the ICDD database (Table 3).

#### 2.1.3. X-ray Phase Analysis of Drugs Containing ACP and Caffeine

Figure 5 and Figure 6 present diffraction patterns of drugs containing caffeine in addition to ACP: Gripex Control, Apap Extra, Panadol Extra, Saridon, and Cefalgin. For all drugs listed above, diffraction lines originating from ACP were identified in the diffraction patterns, consistent with the data in the PDF 00–054–2055 card (Figure 5 and Figure 6). Also, a line originating from caffeine is present in the diffraction patterns (2θ = 11.9174°). It is the strongest diffraction line of caffeine, according to the PDF 00–005–0149 card. This line overlaps one ACP line (2θ = 12.0478°, PDF 00–054–2055), therefore, these two lines are observed as one broadened line in the diffraction pattern (Figure 5 and Figure 6).

|Δ2θ| values for ACP and caffeine are lower than 0.2°, confirming presence of these components in the tested drugs. A comparison of diffraction line intensities for the examined drugs with the ICDD database indicates presence a preferred orientation of grains in the examined samples–experimental line intensities are higher than those reported in the database (Table 4).

#### 2.1.4. X-ray Phase Analysis of Drugs Containing ACP, Caffeine and ASA

Figure 7 summarizes diffraction patterns of the drugs Etopiryna Extra and Excedrin Migra Stop, containing caffeine and ASA apart from ACP. In the diffraction patterns, diffraction lines originating from ACP and lines characteristic for caffeine and ASA are evident. The lines characteristic for ACP are consistent with the PDF 00–027–1902 card. The main diffraction line originating from caffeine occurs at an angle of 2θ = 11.9174° (PDF 00–005–0149) and overlaps the ACP line at an angle of 2θ = 11.9985° (PDF 00–027–1902). A broadened line forms in the diffraction pattern as a result of overlapping of these two diffraction lines. A similar phenomenon is observed for the strongest line of ASA, occurring at an angle of 2θ = 15.5331° (PDF 00–001–0182). This line overlaps the strongest ACP line (2θ = 15.4831°, PDF 00–027–1902).

Intensity of the common ACP-ASA line at an angle of 2θ = 15.4831° indicates a strongly preferred orientation of crystallites of both components or one of them. |Δ2θ| values for all components do not exceed 0.2°, and the values of interplanar distances *d_hkl_* determined for them are close to those from the ICDD database (Table 5).

#### 2.1.5. X-ray Phase Analysis of Drugs Containing ACP, Ascorbic Acid and Other APIs

Figure 8 illustrates diffraction patterns for the drugs Febrisan, Gripex Hot Max, and Fervex. These three drugs have the form of powder for preparation of orally administered solution. The observed diffraction lines originating from ACP are consistent with the PDF 00–054–2055 card for Febrisan and with the PDF 00–027–1902 card for Gripex Hot Max and Fervex. The diffraction lines originating from ACP have low intensities despite a high declared amount of the API (Table 6). It may prove a lower ACP amount in the examined samples than that declared, or a presence of auxiliary substances composed of heavier atoms than ACP and crystallising in high-symmetry systems. High-symmetry phases composed of heavy atoms exhibit higher reflex intensities than phases composed of light atoms. Also, a strongly preferred orientation of crystallites of unidentified auxiliary substances is evident in the tested drugs. No conformity between position of these lines in the diffraction patterns (2θ angles) and the data from the ICDD database has been found. It pertains to such auxiliary substances declared by the manufacturers as, for instance, cellulose, citric acid or sucrose. Also, no diffraction lines originating from ascorbic acid have been identified.

Positions of the experimental diffraction lines for ACP remain in a good accordance with the values of 2θ angles in the ICDD database. |Δ2θ| values are lower than 0.2°. The determined values of interplanar distances *d_hkl_* for the individual lines are close to the data from the ICDD database (Table 6). It confirms the ACP presence in the tested drugs.

#### 2.1.6. X-ray Phase Analysis of Drugs Containing ACP and Other APIs

Figure 9 shows diffraction patterns for the drugs Metafen, Palgotal, Antidol 15, FluControl and Agrypin. This drug group contains also other APIs apart from ACP, such as tramadol, codeine, ibuprofen, pseudoephedrine, and phenylpherine (Table 7). ACP diffraction lines are present in all diffraction patterns, consistent with the PDF 00–054–2055 card. In Metafen, distinct diffraction lines originating from the other API, ibuprofen, are observed. Its amount is relatively large (200 mg).

A comparison of the experimental diffraction data with those from the ICDD database (2θ angles and interplanar distances *d_hkl_*) indicates their very good agreement (Table 8). |Δ2θ| values are lower than 0.2°, confirming presence of ACP in the tested drugs. The observed intensity difference between the experimental diffraction lines and those from the ICDD database indicates occurrence of a preferred orientation of crystallites in the examined samples. For Antidol 15 and Agrypin, the strongest diffraction lines, originating from additional APIs in their composition, overlap with lines originating from ACP (Figure 9, Table 8). A similar phenomenon is observed for Palgotal [26]. The amount of phenylpherine, in FluControl, is about 1% weight. This is close to the value of the limit of detection of crystalline phases and therefore, the lines, originating from phenylpherine are absent on diffraction image.

For the majority of the investigated drugs, no lines originating from auxiliary substances have been identified, meaning that their contents in the tested drugs are small, below the roentgenographic limit of detection. The following drugs are exceptions: Febrisan, Gripex Hot Max, and Fervex. Diffraction lines originating from ACP and auxiliary substances are present in the diffraction patterns of these drugs. These lines are difficult to identify, despite the fact they are included in the database for these substances: citric acid, cellulose, sucrose etc. Possibly, these auxiliary substances interact with one another, forming new substances which are difficult to identify. Another possible cause is presence of such auxiliary substances, for which there are no diffraction data in the ICDD PDF2 database.

### 2.2. Thermal Analysis

Thermal studies using DSC/TGA techniques were carried out for pure ACP and for 15 selected drugs. All preparations were tested under the same conditions. Table 9 gathers thermodynamic data for the examined drugs, obtained by DSC/TGA.

#### 2.2.1. DSC/TGA Measurements for Pure ACP and Drugs Containing Only ACP as API

Figure 10 and Figure 11 show results of thermal analysis of pure ACP and drugs containing only ACP as the API. DSC analysis of pure ACP has shown first endothermal peak caused by fusion of the substance at 172 °C (ΔH = 171.5 J/g). Meanwhile, the second endothermal peak caused by thermal decomposition of ACP occurs at 364 °C (ΔH = 465.4 J/g) (Figure 10). The determined melting point of 172 °C is consistent with the literature data (168–172 °C [27,28,29,30]). Purity of ACP has been determined based on the van’t Hoff equation [23,31]:(3)$$T_{m}=T_{0}-\frac{RT_{0}^{2}X}{\Delta H_{f}}$$
where *T_m_*—sample temperature (in K), *T*_0_—melting temperature of pure ACP (in K), *R*—gas constant (8.314 J/mol·K), *X*—molar fraction of impurities, Δ*H_f_*—heat of fusion (J/mol). Data of pure ACP quoted on NIST website have been used for calculations [32]. Molar fraction of impurities *X* determined from out measurements amounts to 0.084. Crystallinity degree has been determined based on the following formula [33,34]:(4)$$w_{c}=\frac{\Delta H_{f}}{\Delta H_{f}^{0}}$$
where $\Delta H_{f}$—heat of fusion of the tested substance, $\Delta H_{f}^{0}$—heat of fusion of a completely crystalline substance (*w_c_* =100%). The degree of crystallinity calculated from Formula (5) for the studied ACP amounted to *w_c_* = 97%. The TGA/DTG (derivative of thermogravimetry—DTG) curve of pure ACP proves its stability up to the temperature of 172 °C. No loss of weight or dehydration, connected with formation of a residue, has been observed, indicating that the thermal composition is complete (Figure 10). The peak at 172 °C should be attributed to fusion of ACP. Next, at 364 °C, a loss of weight occurs (76%), indicating decomposition of ACP. At this temperature, carbonic matter forming in the main stage of decomposition (172 °C), undergoes further decomposition to gaseous products.

Figure 10 and Figure 11 show DSC and TGA/DTG curves for drugs containing only ACP (Acenol, Paracetamol Biofarm, Paracetamol Aflofarm, Paracetamol Apteo, and Apap USP). For all these drugs, a decrease in the endothermal peak height has been observed in comparison to that of pure ACP (Table 9).

In the group of drugs containing only ACP as the API (Acenol, Paracetamol Biofarm, Paracetamol Aflofarm, Apap USP—Figure 10 and Figure 11), it has been observed that the first endothermal peak, indicating their melting temperature, occurs in the temperature range of 174–176 °C. On the other hand, the second endothermal peak, indicating decomposition of the tested drug, occurs in the temperature range of 359–339 °C. These temperature values are in a good accordance with the data for pure ACP. A slight shift of the second endothermal peak in relation to that of pure ACP (364 °C) confirms presence of a small amount of auxiliary substances in the tested drugs (Figure 10 and Figure 11, Table 9). Paracetamol Apteo (Figure 11) is an exception, as its endothermal peaks are shifted towards lower temperatures: 161 °C and 303 °C, respectively. It may indicate presence of auxiliary substances in amorphous form, which is indicated also by an increased background in the diffraction pattern of Paracetamol Apteo (Figure 4). Loss of weight, determined from TG curves, ranges from 73 to 77% (Table 9).

#### 2.2.2. DSC/TGA Measurements for Drugs Containing ACP and Caffeine

Figure 12 shows DSC and TGA/DTG curves for the drugs containing caffeine in addition to ACP Gripex Control, Cefalgin, Panadol Extra, and Saridon. A decrease in heights of endothermal peaks in comparison to pure ACP is observed in the DSC curves for the examined drugs (Table 9). Simultaneously, for Saridon and Cefalgin, the peaks are shifted towards lower temperatures (Figure 12). This phenomenon indicates that caffeine may be considered an “impurity” of pure ACP in case of these drugs. The ratio of pure ACP to caffeine is 5:1. The fusion process of Saridon and Cefalgin occurs at a lower temperature. On the other hand, the second endothermal peak, determined based on the DSC and DTG curves and indicating decomposition of the tested drugs, appears at higher temperatures: 332 °C and 346 °C, respectively. The loss in weight determined based on the TGA curves amounts to 77% for Saridon and 73% for Cefalgin.

In the case of Gripex Control and Panadol Extra, the first endothermal peak occurs at 165 °C, or at a temperature close to that of pure ACP. It is connected with a higher ACP content in Gripex Control and Panadol Extra. The ratio of pure ACP to caffeine in these drugs is 10:1. Gripex Control undergoes a complete decomposition at 349 °C, and the loss in weight determined based on the TGA curve amounts to 61%. The decomposition process of Panadol Extra, evident as two peaks in the DTG curve, begins at 298 °C. At this temperature, the loss in weight determined based on the TGA curve amounts to approx. 30%. At the second stage, at 352 °C, the loss in weight amounts to 64%. For all tested drugs containing caffeine, no endothermal peak originating from caffeine has been observed. In the case of pure caffeine, an endothermal peak appears at approx. 235–238 °C [35].

#### 2.2.3. DSC/TGA Measurements of Drugs Containing ACP, Caffeine and ASA

Figure 13 shows DSC and TGA/DTG curves for Excedrin Migra Stop and Etopiryna Extra. Both these drugs also contain caffeine and ASA apart from ACP.

A decrease in heights of endothermal peaks, as well as their broadening and a shift towards lower temperatures are observed in the DSC curves (Figure 13, Table 9). This proves a presence of “impurities” in ACP in the form of ASA and caffeine. In Excedrin Migra Stop, the quantitative ratio of ACP to the other APIs is 5:6, while in Etopiryna Extra–2:3. For these two drugs, no endothermal peaks originating from caffeine and ASA are observed. The amounts of ACP and ASA is similar in these drugs, so it may be assumed that the first endothermal peak indicates fusion of the mixture of these components. Melting temperature of ASA amounts to 134–140 °C [36]. Temperature values at which fusion of the tested drug occurs and at which its complete decomposition occurs, depend significantly on the composition of the tested drug (Figure 13). These values, determined from DSC and DTG curves, are shown in Figure 13 and Table 9. Weight losses for Excedrin Migra Stop and Etopiryna Extra, determined based on TGA curves, amount to 67% and 75%, respectively.

#### 2.2.4. DSC/TGA Measurements for Drugs Containing ACP and Other APIs

Figure 14 presents DSC and TGA/DTG curves for Palgotal, Metafen, Agrypin, and FluControl. These drugs contain other APIs (Table 1). Endothermal peaks present in the DSC curves occur at temperatures close to those of pure ACP (Figure 14). Amounts of other APIs present in these drugs are small enough (Table 7) to not affect the positions of endothermal peaks. Metafen is an exception here, being a mixture of ACP (325 mg) and ibuprofen (200 mg). An endothermal peak at 76 °C is present in the DSC curve for Metafen, indicating the presence of ibuprofen. The melting temperature of pure ibuprofen is 75–78 °C [37]. Temperatures at which complete decomposition of the tested drugs occurs, determined from DTG curves, range from 323 to 345 °C. A slight shift towards lower temperatures proves presence of other substances in the drugs, constituting “impurities” of the latter. In the case of FluControl the courses of DSC and TG/DTG curves are the closest to those of pure ACP (Figure 9 and Figure 13). It is caused by the fact that phenylephrine hydrochloride content in the tested drug is the lowest (10 mg) in comparison to the other drugs (Table 7). Loss of weight, determined from TGA curves, ranges from 52 to 72% (Table 9). In the case of drugs containing, among others, ascorbic acid, namely Fervex, Febrisan, and Gripex Hot Max, thermal analysis has not been carried out. These drugs exploded when heated.

## 3. Materials and Methods

### 3.1. Materials

Twenty one drugs containing ACP (acetaminophen/paracetamol) as the main API were selected for the study. Twenty of these drugs are commonly available products which may be purchased in a pharmacy, a shop or via the Internet. One medicament, called Palgotal, is available only by prescription. All the examined drugs are listed in Table 7. In the table, data declared by the manufacturers are taken into account: paracetamol content and presence of other APIs. Pure ACP (Sigma, St. Louis, MO, USA) was used as a reference. The table does not include auxiliary substances such as starch, talc, magnesium stearate, citric acid etc., because of the fact that the main goal of the paper is identification of ACP as the API in the analysed drugs. However, for some of the drugs, diffraction lines originating from other APIs declared by manufacturers were marked, because of their high intensities.

### 3.2. Methods

#### 3.2.1. X-ray Analysis of Samples of Selected Drugs Containing ACP and Pure ACP

Samples of ACP and drugs were ground very thoroughly in an agate mortar, to obtain a fine homogeneous powder. The tests were carried out using polycrystalline diffractometers: D5000 (Siemens, Munich, Germany) and PW3050 (X’Pert Philips, Malvern Panalytical, Malvern, UK). Bragg-Brentano focusing of diffractive radiation was applied. Total analysis time of every sample amounted to approx. 2 h. Operating conditions of the diffractometers are presented in Table 10.

Because of the fact that in the obtained diffraction patterns, the strongest diffraction lines are observed at low angles, all diffraction images are shown for a 2θ angular range of 10°–40°. For a comparative analysis of the experimental data with the standards, diffractometric data from the ICDD PDF2 database Release 2008 were used. The numbers of the PDF cards, used to identification of diffraction lines, are gathered in Table 11.

#### 3.2.2. Thermal Analysis

Proper weighed amounts of pure ACP and of the examined drugs were placed in aluminium crucibles with lids and heated to a temperature of 450 °C under argon, with a heating rate of 5 °C/min. The measurements were carried out using a LabsysEvo device (Setaram, Caluire-et-Cuire, France). All measurement conditions were used for all studied drugs.

## 4. Conclusions

Pure acetaminophen and 21 generally available medicinal drugs containing ACP were tested using X-ray radiation. Identification of ACP in the examined drugs was performed using X-ray phase analysis, based on a comparison of positions of diffraction lines (2θ angles) for the tested samples, their intensities, and interplanar distances *d_hkl_* with the data from the ICDD PDF2 database. The tests carried out allowed for ascertaining that ACP is present in the tested drug samples, as per manufacturer’s declaration. In most cases, the difference of 2θ angle values between the experimental data and those from the database is smaller than 0.2°. It indicates use of substances with the same structural parameters or substances being the same crystalline varieties. It may be confirmed that ACP present in the compositions of the investigated drugs is a form crystallising in monoclinic system (form I). For the majority of the studied drugs, no diffraction lines originating from auxiliary substances were identified, proving presence of small amounts of these substances. Moreover, pure ACP and 15 selected drugs were subjected to DSC/TGA tests. In the case of pure ACP, molar fraction of impurities and degree of crystallinity were determined. An analysis of DSC and TGA curves confirmed presence of ACP in the tested drugs. Melting temperatures of drugs containing only ACP as the API are close to that of pure ACP, which confirms small amounts of auxiliary substances in the tested drugs, having no impact on the determined thermal parameters. A broadening of endothermal peaks in the DSC curves and shifts of the peaks towards lower temperatures together with an increase in the amount of substances other than ACP, were observed. Presence of other APIs and auxiliary substances acts as “impurity” of the main substance, leading to changes in shape and positions of endothermal peaks. On the basis of the tests carried out by thermoanalytical and X-ray techniques, it may be ascertained that a combination of both methods (XRPD and DSC/TGA) can be utilised for distinguishing between original drugs from counterfeit products, for instance by checking the presence of the proper API or its adequate polymorphic form. Determination of behaviour of the tested drugs at various temperatures allows for defining the temperature ranges below which the analysed substances may be processed without changing their physicochemical properties. The obtained results may be useful also as an indication for stability tests of various drugs and can serve the purpose of detection of inconsistencies in compositions of drugs. In such a case, when the results of diffractometric analysis and thermal analysis are ambiguous or raise some suspicions as for authenticity of the product, the tests should be repeated or the analysis extended to other solid-phase analytical methods or separation methods, such as chromatography, electrophoresis etc.

## Figures and Tables

**Figure 1 molecules-25-05909-f001:**
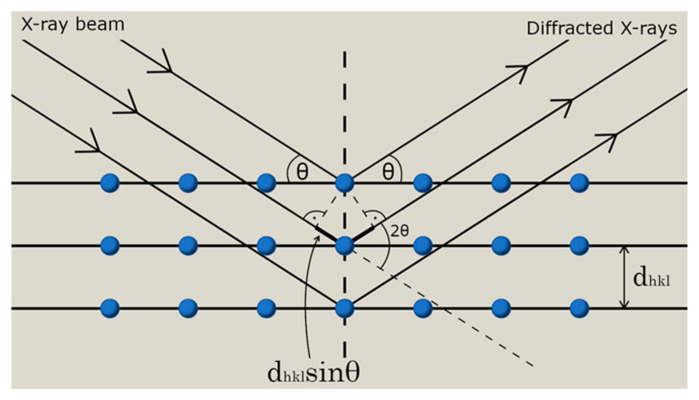
X-ray diffraction in a set of parallel planes (*hkl*) in crystal.

**Figure 2 molecules-25-05909-f002:**
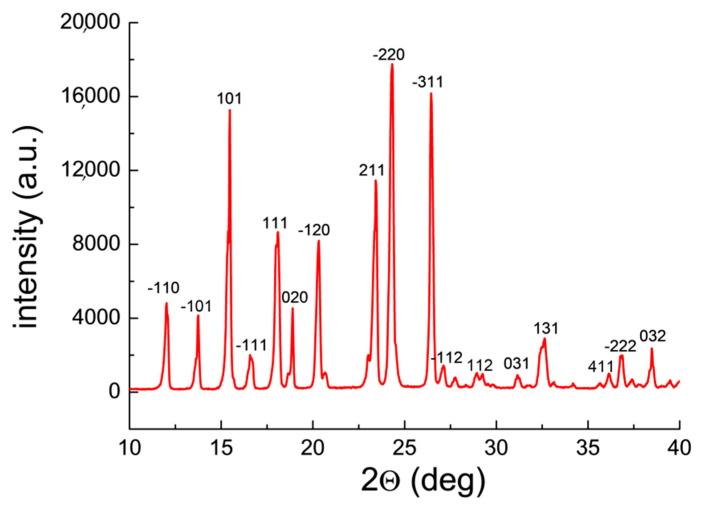
Diffraction pattern of pure acetaminophen API.

**Figure 3 molecules-25-05909-f003:**
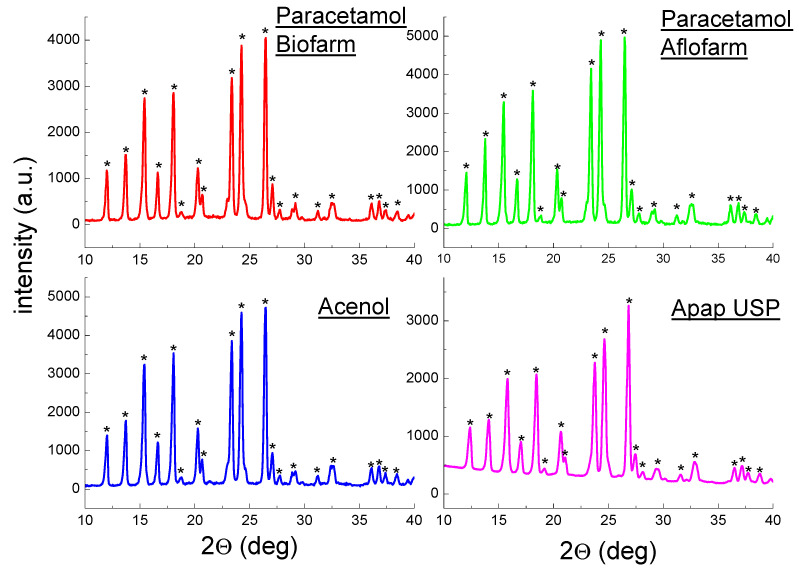
Diffraction patterns for drugs: Paracetamol Biofarm, Paracetamol Aflofarm, Acenol and Apap USP. (*—acetaminophen).

**Figure 4 molecules-25-05909-f004:**
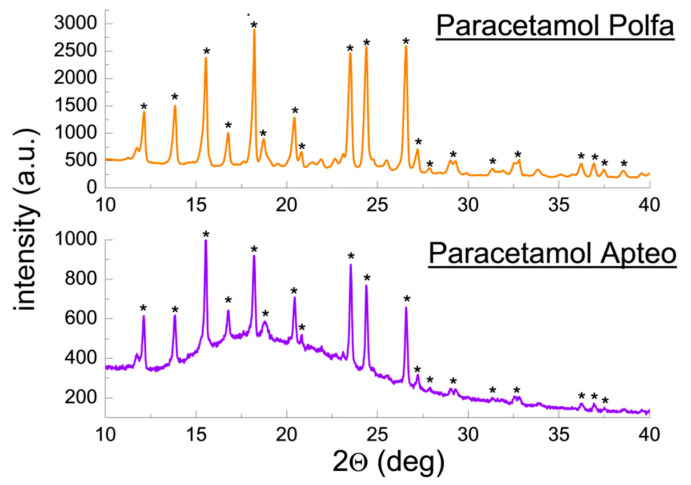
Diffraction patterns for the drugs Paracetamol Polfa and Paracetamol Apteo. (*—acetaminophen).

**Figure 5 molecules-25-05909-f005:**
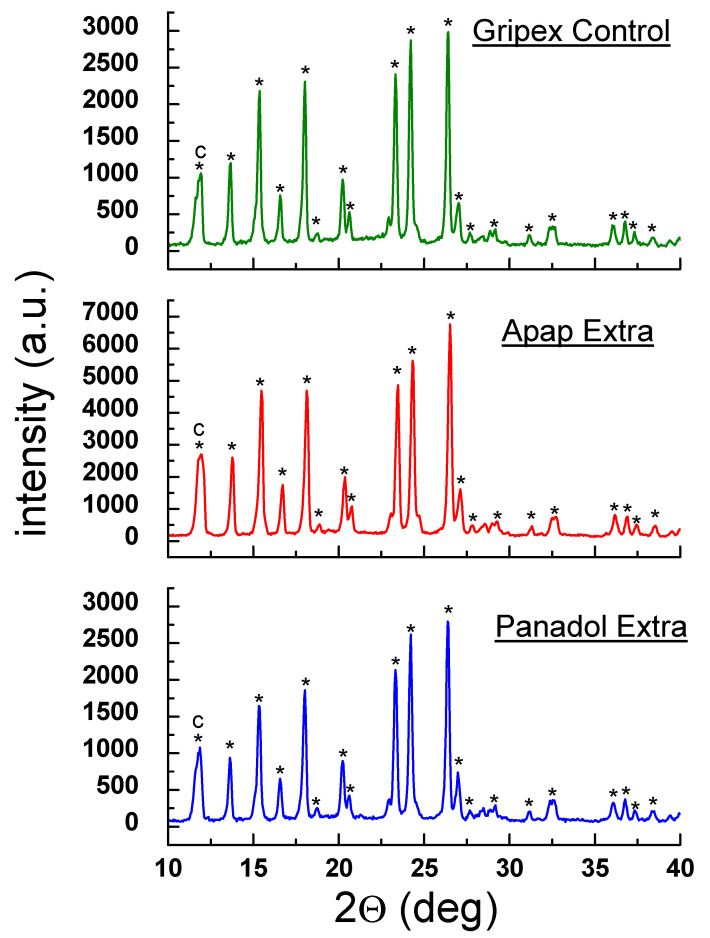
Diffraction patterns for drugs: Gripex Control, Apap Extra, and Panadol Extra. (*—acetaminophen, c—caffeine).

**Figure 6 molecules-25-05909-f006:**
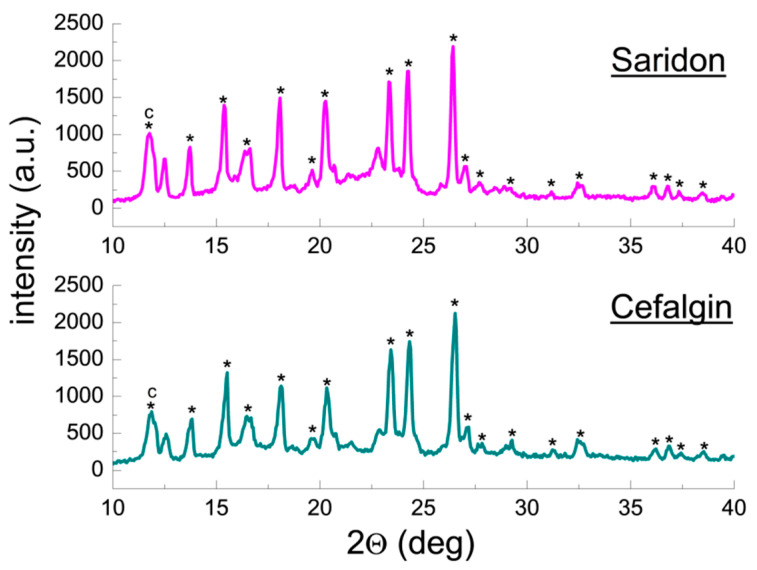
Diffraction patterns for drugs: Saridon and Cefalgin. (*—acetaminophen, c—caffeine).

**Figure 7 molecules-25-05909-f007:**
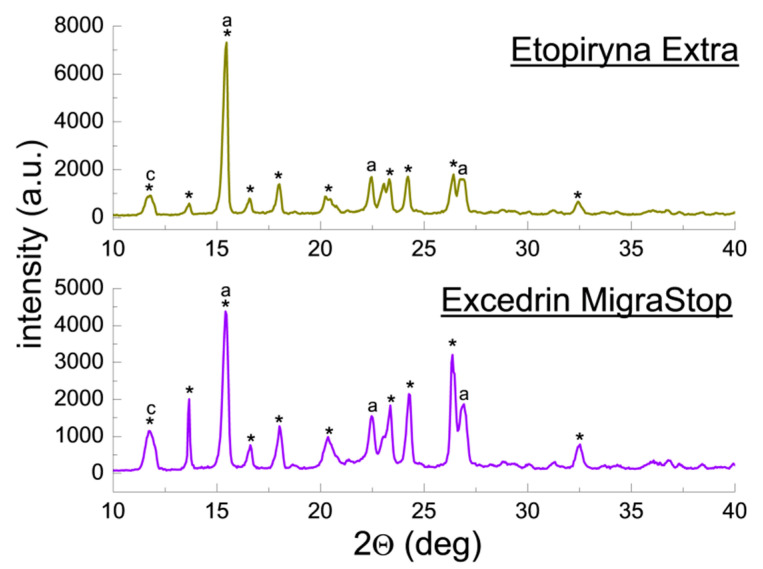
Diffraction patterns for the drugs Etopiryna Extra and Excedrin Migra Stop. (*—acetaminophen, a—acetylsalicylic acid, c—caffeine).

**Figure 8 molecules-25-05909-f008:**
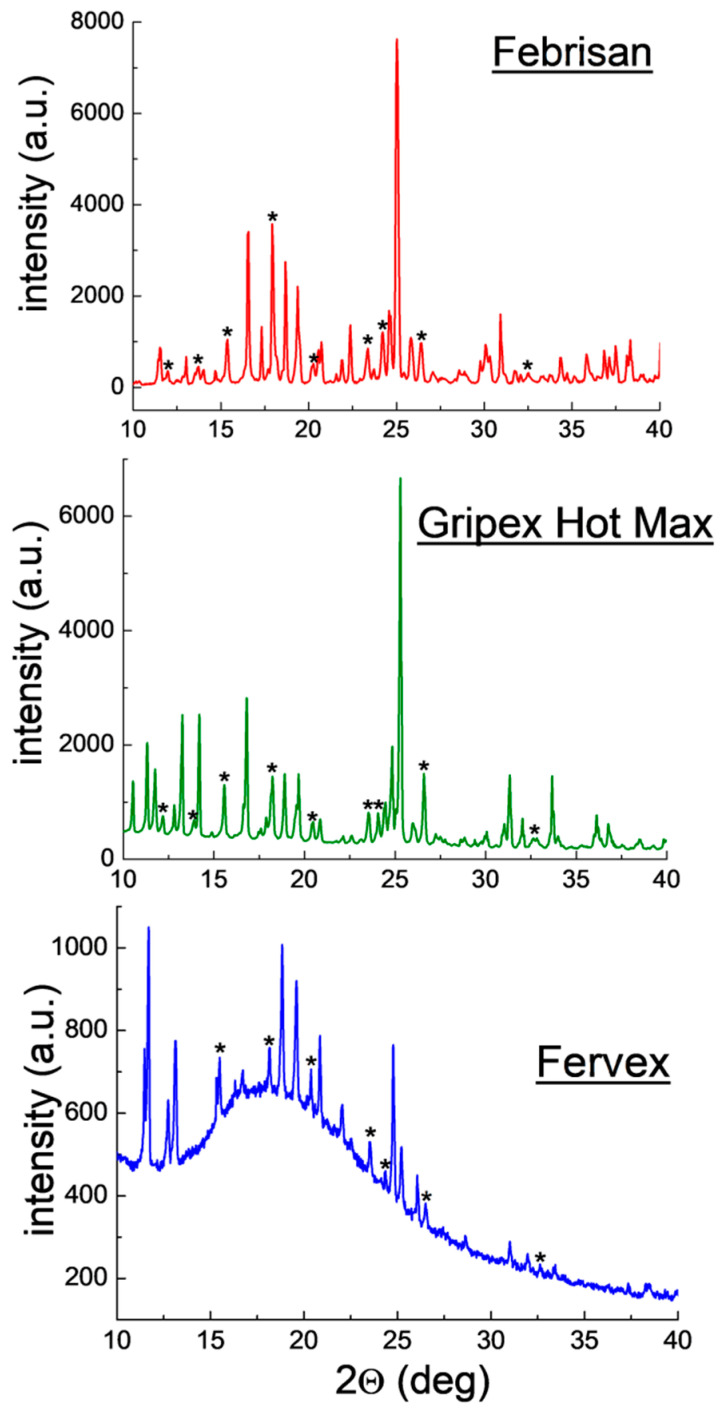
Diffraction patterns for the drugs Febrisan, Gripex Hot Max and Fervex. (*—acetaminophen).

**Figure 9 molecules-25-05909-f009:**
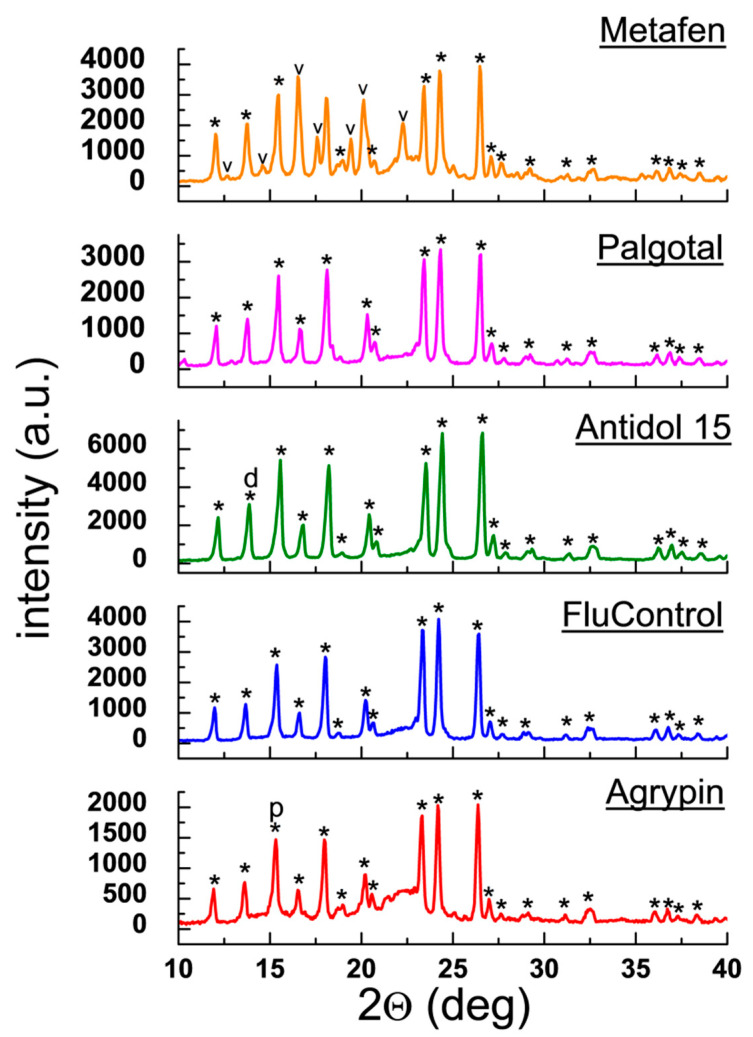
Diffraction patterns for the drugs Metafen, Palgotal, Antidol 15, FluControl and Agrypin. (*—acetaminophen, ∨—ibuprofen, d—codeine phosphate and p—pseudoephedrine hydrochloride).

**Figure 10 molecules-25-05909-f010:**
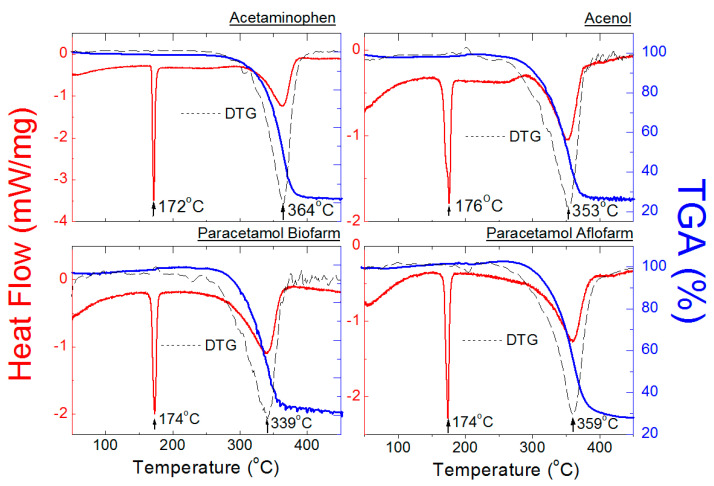
DSC and TGA/DTG curves of ACP and the drugs containing ACP Acenol, Paracetamol Biofarm, Paracetamol Aflofarm.

**Figure 11 molecules-25-05909-f011:**
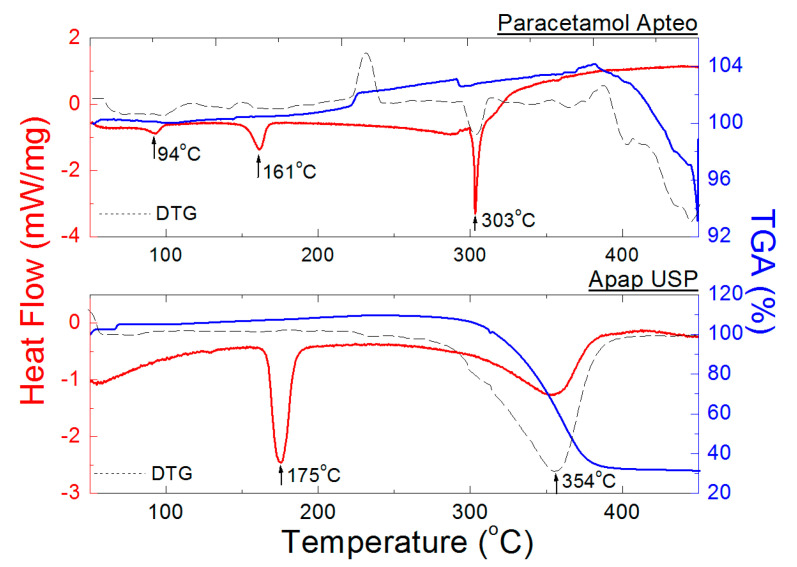
DSC and TGA/DTG curves of the drugs containing ACP Paracetamol Apteo and Apap USP.

**Figure 12 molecules-25-05909-f012:**
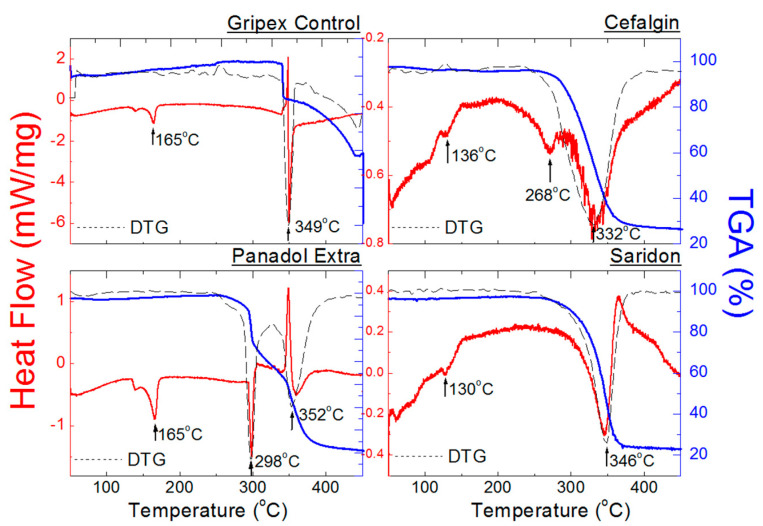
DSC and TGA/DTG curves of the drugs containing ACP and caffeine Gripex Control, Cefalgin, Panadol Extra, and Saridon.

**Figure 13 molecules-25-05909-f013:**
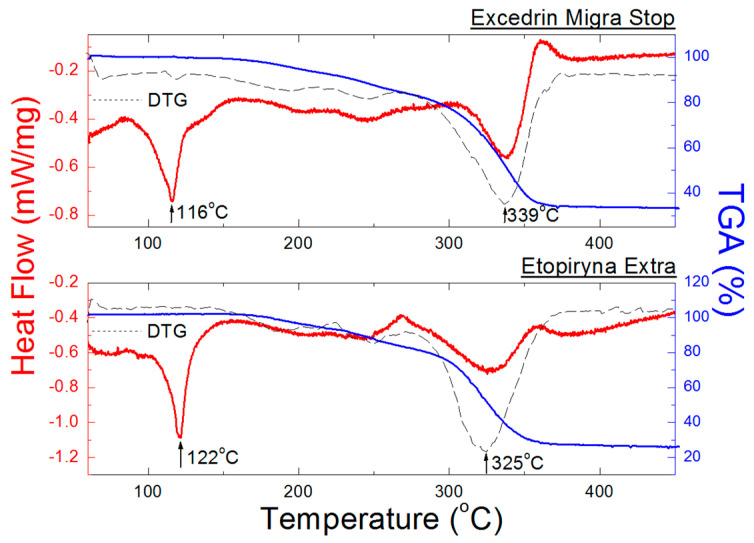
DSC and TGA/DTG curves of the drugs containing ACP, caffeine and ASA Excedrin Migra Stop and Etopiryna Extra.

**Figure 14 molecules-25-05909-f014:**
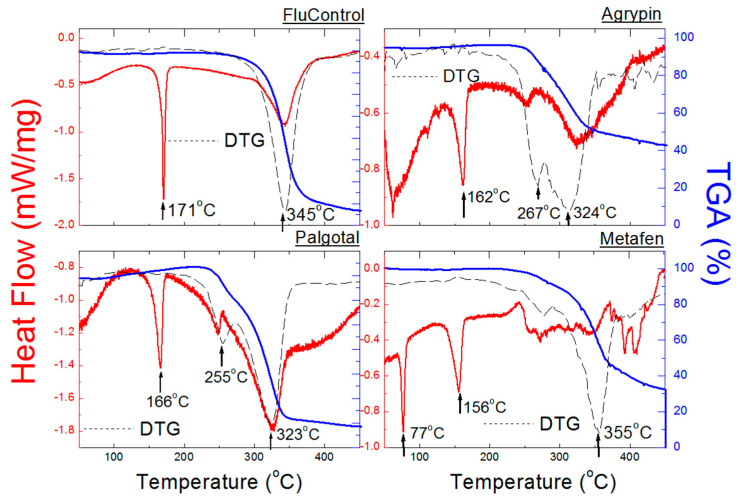
DSC and TGA/DTG curves of the drugs Palgotal, Metafen, Agrypin, and FluControl containing ACP and other APIs.

**Table 1 molecules-25-05909-t001:** Comparison of experimental data with data from ICDD database for pure ACP (C_8_H_9_NO_2_).

No. of Diffraction Line	2θ (°) Exp.	2θ (°) ICDD	J/J_max_ (%) Exp.	J/J_max_ (%) ICDD	|Δ2θ|	*d_hkl_* (Å) Exp.	*d_hkl_* (Å) ICDD
Acetaminophen (C_8_H_9_NO_2_ PDF 00–039–1503)							
1	15.4853	15.5166	87	71	0.0313	5.7175	5.706
2	18.0840	18.1936	50	56	0.1096	4.9013	4.872
3	23.4478	23.4907	65	59	0.0429	3.7908	3.784
4	24.3119	24.3729	100	100	0.0610	3.6580	3.649
5	26.4561	26.5300	92	77	0.0739	3.3662	3.357

**Table 2 molecules-25-05909-t002:** Comparison of experimental data with data from the ICDD database for the drugs Acenol, Apap USP, Paracetamol Biofarm and Paracetamol Aflofarm containing only ACP as API.

No. of Diffraction Line	2θ (°) Exp.	2θ (°) ICDD	J/J_max_ (%) Exp.	J/J_max_ (%) ICDD	|Δ2θ|	*d_hkl_* (Å) Exp.	*d_hkl_* (Å) ICDD
Acetaminophen (C_8_H_9_NO_2_ PDF 00–054–2055)							
**Acenol**							
1	26.4562	26.4258	100	100	0.0304	3.3662	3.37
2	24.2416	24.2313	98	80	0.0103	3.6685	3.67
3	23.3774	23.3904	83	66	0.0130	3.8021	3.80
**Apap USP**							
1	26.4581	26.4258	100	100	0.0323	3.3660	3.37
2	24.2411	24.2313	84	80	0.0098	3.6685	3.67
3	23.3762	23.3904	72	66	0.0142	3.8023	3.80
**Paracetamol Biofarm**							
1	26.4561	26.4258	100	100	0.0303	3.3662	3.37
2	24.2735	24.2313	95	80	0.0422	3.6637	3.67
3	23.3774	23.3904	78	66	0.0130	3.8021	3.80
**Paracetamol Aflofarm**							
1	26.4906	26.4258	100	100	0.0648	3.3619	3.37
2	24.3326	24.2313	98	80	0.1013	3.6549	3.67
3	23.3971	23.3904	85	66	0.0067	3.7989	3.80

**Table 3 molecules-25-05909-t003:** Comparison of experimental data with data from ICDD database for the drugs Paracetamol Polfa and Paracetamol Apteo containing only ACP as API.

No. of Diffraction Line	2θ (°) Exp.	2θ (°) ICDD	J/J_max_ (%) Exp.	J/J_max_ (%) ICDD	|Δ2θ|	*d_hkl_* (Å) Exp.	*d_hkl_* (Å) ICDD
Acetaminophen (C_8_H_9_NO_2_ PDF)							
**Paracetamol Polfa**							
1	15.5471	15.4512	84	100	0.0959	5.6949	5.73
2	18.2247	18.1636	100	95	0.0611	4.8638	4.90
3	23.5099	23.3281	86	85	0.1818	3.7824	3.80
**Paracetamol Apteo**							
1	15.5180	15.4512	100	100	0.0668	5.7055	5.73
2	18.1898	18.1636	94	95	0.0262	4.8730	4.90
3	23.5099	23.3281	89	85	0.1815	3.7824	3.80

**Table 4 molecules-25-05909-t004:** Comparison of experimental data with data from the ICDD database for the drugs Gripex Control, Apap Extra, Panadol Extra, Saridon, and Cefalgin containing ACP (C_8_H_9_NO_2_) and caffeine (C_8_H_10_N_4_O_2_).

No. of Diffraction Line	2θ (°) Exp.	2θ (°) ICDD	J/J_max_ (%) Exp.	J/J_max_ (%) ICDD	|Δ2θ|	*d_hkl_* (Å) Exp.	*d_hkl_* (Å) ICDD
**Gripex Control**							
Acetaminophen (C_8_H_9_NO_2_ PDF 00–054–2055)							
1	26.3955	26.4258	100	100	0.0303	3.3738	3.37
2	24.2077	24.2313	96	80	0.0236	3.6735	3.67
3	23.3163	23.3904	83	66	0.0768	3.8119	3.80
Caffeine (C_8_H_10_N_4_O_2_ PDF 00–005–0149)							
1	11.8817	11.9174	37	100	0.0357	7.4422	7.42
**Apap Extra**							
Acetaminophen (C_8_H_9_NO_2_ PDF 00–054–2055)							
1	26.5306	26.4258	100	100	0.1048	3.3569	3.37
2	24.3473	24.2313	84	80	0.1160	3.6528	3.67
3	23.4514	23.3904	74	66	0.0610	3.7903	3.80
Caffeine (C_8_H_10_N_4_O_2_ PDF 00–005–0149)							
1	11.8817	11.9174	43	100	0.0357	7.4422	7.42
**Panadol Extra**							
Acetaminophen (C_8_H_9_NO_2_ PDF 00–054–2055)							
1	26.3956	26.4258	100	100	0.0302	3.3738	3.37
2	24.2077	24.2313	96	80	0.0236	3.6735	3.67
3	23.3164	23.3904	80	66	0.0740	3.8119	3.80
Caffeine (C_8_H_10_N_4_O_2_ PDF 00–005–0149)							
1	11.8187	11.9174	40	100	0.0987	7.4817	7.42
**Saridon**							
Acetaminophen (C_8_H_9_NO_2_ PDF 00–054–2055)							
1	26.4377	26.4258	100	100	0.0119	3.3685	3.37
2	24.2666	24.2313	86	80	0.0353	3.6647	3.67
3	23.3527	23.3904	79	66	0.0377	3.8060	3.80
Caffeine (C_8_H_10_N_4_O_2_ PDF 00–005–0149)							
1.	11.7695	11.9174	49	100	0.1479	7.5128	7.42
**Cefalgin**							
Acetaminophen (C_8_H_9_NO_2_ PDF 00–054–2055)							
1.	26.5308	26.4258	100	100	0.1050	3.3570	3.37
2.	24.3306	24.2313	84	80	0.0993	3.6552	3.67
3.	23.4168	23.3904	79	66	0.0264	3.7958	3.80
Caffeine (C_8_H_10_N_4_O_2_ PDF 00–005–0149)							
1.	11.8335	11.9174	39	100	0.0839	7.4724	7.42

**Table 5 molecules-25-05909-t005:** Comparison of experimental data with data from the ICDD database for the drugs Etopiryna Extra and Excedrin MigraStop containing ACP (C_8_H_9_NO_2_), ASA (C_9_H_8_O_4_) and caffeine (C_8_H_10_N_4_O_2_).

No. of Diffraction Line	2θ (°) Exp.	2θ (°) ICDD	J/J_max_ (%) Exp.	J/J_max_ (%) ICDD	|Δ2θ|	*d_hkl_* (Å) Exp.	*d_hkl_* (Å) ICDD
**Etopiryna Extra**							
Acetaminophen (C_8_H_9_NO_2_ PDF 00–027–1902)							
1	15.4831	15.4512	100	100	0.0319	5.7183	5.73
2	18.0326	18.1636	20	95	0.1310	4.9151	4.90
3	23.3876	23.3281	27	85	0.0595	3.8004	3.80
Acetylsalicylic acid (C_9_H_8_O_4_ PDF 00–001–0182)							
1	15.4831	15.5331	100	100	0.0500	5.7183	5.70
2	26.8451	26.9968	25	40	0.1517	3.3183	3.30
3	22.4738	22.6647	27	32	0.1908	3.9529	3.92
Caffeine (C_8_H_10_N_4_O_2_ PDF 00–005–0149)							
1	11.7695	11.9174	15	100	0.1479	7.5129	7.42
**Excedrin MigraStop**							
Acetaminophen (C_8_H_9_NO_2_ PDF 00–027–1902)							
1	15.4249	15.4512	100	100	0.0263	5.7397	5.73
2	18.0326	18.1636	33	95	0.1310	4.9151	4.90
3	23.3877	23.3281	45	85	0.0596	3.8004	3.80
Acetylsalicylic acid (C_9_H_8_O_4_ PDF 00–001–0182)							
1	15.4249	15.5331	100	100	0.1082	5.7397	5.70
2	26.9092	26.9968	45	40	0.0876	3.3105	3.30
3	22.4738	22.6647	39	32	0.1908	3.9529	3.92
Caffeine (C_8_H_10_N_4_O_2_ PDF 00–005–0149)							
1	11.7404	11.9174	30	100	0.1770	7.5314	7.42

**Table 6 molecules-25-05909-t006:** Comparison of experimental data with data from the ICDD database for the drugs Febrisan, Gripex Hot Max, and Fervex.

No. of Diffraction Line	2θ (°) Exp.	2θ (°) ICDD	J/J_max_ (%) Exp.	J/J_max_ (%) ICDD	|Δ2θ|	*d_hkl_* (Å) Exp.	*d_hkl_* (Å) ICDD
Acetaminophen (C_8_H_9_NO_2_ PDF 00–054–2055)							
**Febrisan**							
1	26.4287	26.4258	17	100	0.0029	3.3696	3.37
2	24.1954	24.2313	20	80	0.0359	3.6754	3.67
3	23.3222	23.3904	15	66	0.0130	3.8110	3.80
Acetaminophen (C_8_H_9_NO_2_ PDF 00–027–1902)							
**Gripex Hot Max**							
1	15.5010	15.4512	21	100	0.0498	5.7117	5.73
2	18.1331	18.1636	23	95	0.0305	4.8881	4.90
3	23.4595	23.3281	14	85	0.1314	3.7890	3.80
**Fervex**							
1	15.5147	15.4512	71	100	0.0635	5.7067	5.73
2	18.1930	18.1636	73	95	0.0294	4.8722	4.90
3	23.5241	23.3281	51	85	0.1960	3.7787	3.80

**Table 7 molecules-25-05909-t007:** Analysed drugs containing ACP and pure ACP as a chemical reagent.

No.	Product Name (*Manufacturer*)	ACP Content in 1 Tablet/Sachet [mg]	Selected Additional APIs
1.	Acetaminophen *(Sigma)*	-	-
2.	Apap (*USP Zdrowie*)	500	-
3.	Paracetamol (*Aflofarm*)	500	-
4.	Acenol (*Galena*)	300	-
5.	Paracetamol (*Biofarm*)	1000	-
6.	Paracetamol (*Polfa Łódź*)	500	-
7.	Paracetamol Apteo (*Synoptis Pharma*)	500	-
8.	Apap Extra (*US Pharmacia*)	500	Caffeine (65 mg)
9.	Gripex Control (*US Pharmacia*)	500	Caffeine (50 mg)
10.	Panadol Extra (*GSK Consumer Healthcare*)	500	Caffeine (65 mg)
11.	Cefalgin (*Adamed*)	250	Caffeine (50 mg)
12.	Saridon (*Bayer*)	250	Caffeine (50 mg)
13.	Excedrin Migra Stop (*GSK Consumer Healthcare*)	250	Caffeine (65 mg) ASA (250 mg)
14.	Etopiryna Extra (*Polpharma*)	200	Caffeine (50 mg) ASA (250 mg)
15.	FluControl (*Aflofarm*)	650	Phenylephrine·HCl (10 mg)
16.	Agrypin (*Polfa Tarchomin*)	325	Pseudoephedrine·HCl (30 mg)
17.	Metafen (*Polpharma*)	325	Ibuprofen (200 mg)
18.	Antidol 15 (*Sandoz*)	500	Codeine phosphate (15 mg)
19.	Palgotal (*Zentiva a Sanofi Company*)	650	Tramadol·HCl (75 mg)
20.	Febrisan (*Takeda Pharma*)	750	Phenylephrine·HCl (10 mg) Ascorbic acid (60 mg)
21.	Gripex Hot Max (*US Pharmacia*)	1000	Phenylephrine·HCl (12.2 mg) Ascorbic acid (100 mg)
22.	Fervex (*Bristol Meyers Sqibb Poland*)	500	Pheniramine maleate (25 mg) Ascorbic acid (200 mg)

**Table 8 molecules-25-05909-t008:** Comparison of experimental data with data from the ICDD database for the drugs Metafen, Palgotal, Antidol 15, FluControl, and Agrypin.

No. of Diffraction Line	2θ (°) Exp.	2θ (°) ICDD	J/J_max_ (%) Exp.	J/J_max_ (%) ICDD	|Δ2θ|	*d_hkl_* (Å) Exp.	*d_hkl_* (Å) ICDD
Acetaminophen (C_8_H_9_NO_2_ PDF 00–054–2055)							
**Metafen**							
1.	26.4545	26.4258	100	100	0.0287	3.3664	3.37
2.	24.2917	24.2313	96	80	0.0604	3.6610	3.67
3.	23.4191	23.3904	84	66	0.0287	3.7954	3.80
Ibuprofen (C_13_H_18_O_2_ PDF 00–030–1757)							
1.	16.5495	16.4941	100	100	0.0554	5.3533	5.37
2.	20.1826	20.1647	79	95	0.0179	4.3961	4.40
3.	17.5587	17.5123	46	60	0.0464	5.0467	5.06
**Palgotal**							
1.	26.4545	26.4258	96	100	0.0287	3.3664	3.37
2.	24.3676	24.2313	100	80	0.1363	3.6454	3.67
3.	23.4191	23.3904	93	66	0.0287	3.7954	3.80
**Antidol 15**							
1.	26.5836	26.4258	100	100	0.1578	3.3503	3.37
2.	24.3678	24.2313	99	80	0.1365	3.6500	3.67
3.	23.4823	23.3904	75	66	0.0919	3.7853	3.80
Codeine phosphate (C_36_H_48_N_2_O_14_P_2_ PDF 00–51–1963)							
1.	14.0459	13.9395	46	100	0.1064	6.3000	6.35
**FluControl**							
1.	26.4158	26.4258	88	100	0.0100	3.3712	3.37
2.	24.2578	24.2313	100	80	0.0265	3.6660	3.67
3.	23.3098	23.3904	91	66	0.0806	3.8130	3.80
**Agrypin**							
1.	26.3534	26.4258	100	100	0.0724	3.3791	3.37
2.	24.1954	24.2313	98	80	0.0359	3.6754	3.67
3.	23.3222	23.3904	91	66	0.0682	3.8110	3.80
Pseudoephedrine hydrochloride (C_10_H_15_NO·HCl PDF 00–029–1885)							
1.	15.3829	15.4512	71	100	0.0683	5.76	5.73

**Table 9 molecules-25-05909-t009:** Parameters determined from TGA and DSC analysis for pure ACP and investigated drugs.

Drug Name	Weight Loss (%)	Onset (°C)	Endset (°C)	Peak Maximum (°C)	Peak Height (mW)	Peak Area (J)	Enthalpy (J/g)
Acetaminophen	76	169 321	175 378	172 364	31.9 10.9	1.715 4.654	171.5 465.4
Apap USP	77	167 238	184 373	175 354	11.4 5.66	0.859 1.830	143.2 305.2
Paracetamol Aflofarm	75	169 306	177 373	174 359	12.5 5.53	1.086 2.795	136.9 430.0
Paracetamol Biofarm	78	168 359	177 372	174 339	13.2 6.97	1.036 3.887	141.9 532.5
Paracetamol Apteo	74	92 155 302	96 167 306	94 161 303	1.68 5.79 19.8	0.185 0.649 0.834	25.74 90.09 115.9
Acenol	73	171 333	180 365	176 353	12.5 7.23	1.325 3.232	154.1 375.8
Gripex Control	61	160	169	165	3.53	0.433	103.1
Cefalgin	73	131 255 289	139 280 337	136 268 332	0.37 0.51 2.42	0.035 0.069 0.730	4.406 8.684 92.38
Saridon	77	129 305	145 358	130 346	0.49 4.99	0.062 1.920	7.936 246.2
Panadol Extra	64	157 294	170 301	165 298	6.23 12.86	1.073 0.802	114.2 85.31
Excedrin Migra Stop	67	99 319	122 354	116 339	4.42 4.56	0.994 1.502	84.25 127.3
Etopiryna Extra	75	116 322	127 330	122 325	5.06 2.65	0.503 0.891	83.83 148.5
Metafen	68	73 148 390 403	80 162 395 421	77 156 393 410	4.77 3.87 1.74 1.98	0.308 0.574 0.114 0.252	31.42 58.53 11.67 25.75
FluControl	72	168 333	175 353	171 345	20.3 9.16	1.562 4.126	109.2 288.5
Agrypin	52	152 254 291	167 278 347	162 267 324	1.95 1.024 1.50	0.280 0.092 1.157	46.66 15.36 192.8
Palgotal	65	152 276	171 353	166 323	2.77 2.74	0.592 2.251	74.02 281.4

**Table 10 molecules-25-05909-t010:** Operational parameters of SIEMENS D5000 and X’Pert Philips (PW3050) diffractometers.

Goniometer	System θ-2θ
Angular range of goniometer	10°–135°
Angular step	0.02°
Anode	Cu
Wavelength	1.54056 Å
Filter	Ni
Voltage	30 kV
Current intensity	30 mA

**Table 11 molecules-25-05909-t011:** List of used ICDD PDF2 (Release 2008) cards.

No.	Chemical Compound	Chemical Formula	No. PDF Card
1.	ACP	C_8_H_9_NO_2_	00–039–1503
2.	ACP	C_8_H_9_NO_2_	00–054–2055
3.	ACP	C_8_H_9_NO_2_	00–027–1902
4.	ACP	C_8_H_9_NO_2_	00–029–1505
5.	ACP	C_8_H_9_NO_2_	00–054–2064
6.	ASA	C_9_H_8_O_4_	00–001–0182
7.	Caffeine	C_8_H_10_N_4_O_2_	00–005–0149
8.	Ibuprofen	C_13_H_18_O_2_	00–030–1757
9.	Ibuprofen	C_13_H_18_O_2_	00–032–1723
10.	Ibuprofen	C_13_H_18_O_2_	00–034–1728
11.	Codeine phosphate	C_36_H_48_N_2_O_14_P_2_	00–51–1963
12.	Pseudoephedrine·HCl	C_10_H_15_NO·HCl	00–029–1885
13.	Pseudoephedrine·HCl	C_10_H_15_NO·HCl	00–031–1858
14.	Pseudoephedrine·HCl	C_10_H_15_NO·HCl	00–041–1946

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
