# Peer review of "X-ray and Thermal Analysis of Selected Drugs Containing Acetaminophen"

_molecules, 2020, doi:10.3390/molecules25245909_

Round 1

Reviewer 1 Report

The authors present a paper concerning the analysis of selected Drugs containing Acetaminophen(paracetamol)

Acetaminophen occurs in two polymorphic variations: monoclinic (form I) and orthorhombic (form II). At room temperature, the form I is more stable thermodynamically than form II. Moreover, This compound is included in the composition of more than 300 pharmaceutical

As the Mass counterfeiting of pharmaceutical products has become the plague of the 21st century, the control of active ingredients is a current challenge for safe drugs and is urgently needed

The authors use two methodologies: X-ray Structural Analysis  and Thermal Analysis

The authors have shown that the identification of ACP in the examined drugs could be performed by using X-ray phase analysis. Analysis of DSC and TGA curves confirmed the presence of ACP in the tested drugs.

The authors have shown that the identification of ACP in the examined drugs could be performed by using X-ray phase analysis. Analysis of DSC and TGA curves confirmed the presence of ACP in the tested drugs.

The authors have shown that it may be ascertained that a combination of both methods (XRPD and DSC/TGA) can be utilized for distinguishing between original drugs from counterfeit products, for instance by checking the presence of the proper API or its adequate polymorphic form.

 Determination of behavior of the tested drugs at various temperatures allows for defining the temperature ranges below which the analyzed substances may be processed without changing their physicochemical properties. The obtained results may be useful also as an indication for stability tests of various drugs and can serve the purpose of detecting inconsistencies in compositions of drugs. In such a case, when the results of diffractometric analysis and thermal analysis are ambiguous or raise some suspicions as to the authenticity of the product, the tests should be repeated or the analysis expanded with other methods, e.g. IR, UV-Vis, chromatography, or microscopy.

In conclusion, I recommend this paper for publication after considering the minor corrections cited below :

  • To complete the introduction, please add some words about the place of counterfeiting in quantitative terms (in general and in the specific cases cited in the text)

Author Response

We thank the Reviewer1 very much for their careful reading of the paper and their constructive remarks. In order to take into account the latter, the paper has been revised. All changes are marked in green.

 Review 1

In conclusion, I recommend this paper for publication after considering the minor corrections cited below :

To complete the introduction, please add some words about the place of counterfeiting in quantitative terms (in general and in the specific cases cited in the text)

Response: A paragraph has been added to the Introduction on countries that are major producers of counterfeit medicines traded worldwide and that are major transit points for counterfeit medicines. The percentage of counterfeit drug classes was entered.

Reviewer 2 Report

Izabela Jendrzejewska and colleagues submitted a paper concerning the physico-chemical analysis of drugs containing acetaminophen. The compound contained in several different pharmaceutical preparations has been analyzed using diffraction and TGA. The data are well presented and overall quality of presentation is very high.

Concerning the experimental design, the authors mainly used two analytical techniques (X-rays and thermal analysis) to investigate this drug. The resulting data are also paralleled by melting point analysis. I wonder if using HPLC analysis (in parallel or as a control) would have been more representative and effective in verifying the identity/purity of the analyte. In particular, I am concerned about the sentence “melting point close to that of pure acetaminophen”. Is it anyhow indicative of identity or purity of the drug? The authors should at least comment on these points.

Introduction is slightly too long. There is in fact a long part concerning general X-ray techniques (e.g. Figure 1) and also the part about TGA is way too specific and not completely focused on the topic of the paper.

In Figure 2, I would suggest using the symbol instead of “theta”.

Conclusions are well written. I suggest to resume and comment more widely on the effect of the presence of other APIs that may interfere with the used analyses.

Author Response

We thank the Reviewer2 very much for their careful reading of the paper and their constructive remarks. In order to take into account the latter, the paper has been revised. All changes are marked in green.

Review 2

I wonder if using HPLC analysis (in parallel or as a control) would have been more representative and effective in verifying the identity/purity of the analyte. In particular, I am concerned about the sentence “melting point close to that of pure acetaminophen”. Is it anyhow indicative of identity or purity of the drug? The authors should at least comment on these points.

Response: Chromatographic methods can be used to analyze counterfeit drugs, although spectroscopic methods are preferred for their speed and ability to analyze untreated samples. Some references on HPLC analysis of acetaminophen have been added.

The sentence "melting point close to that of pure acetaminophen" in the abstract means that the melting points were practically identical to the acetaminophen standard. However, it is always necessary to take into account the measurement error and the influence of other co-formulated APIs or excipients.

Introduction is slightly too long. There is in fact a long part concerning general X-ray techniques (e.g. Figure 1) and also the part about TGA is way too specific and not completely focused on the topic of the paper.

Response: The introduction has been modified and shortened.

In Figure 2, I would suggest using the symbol instead of “theta”.

Response: All figures have been modified.

Conclusions are well written. I suggest to resume and comment more widely on the effect of the presence of other APIs that may interfere with the used analyses. I suggest to resume and comment more widely on the effect of the presence of other APIs that may interfere with the used analyses.

Response: The conclusion has been extended.